

# Comparison of rule- and ordinary differential equation-based dynamic model of DARPP-32 signalling network

Emilia M. Wysocka[1], Matthew Page[2], James Snowden[2] and
T. Ian Simpson[1]

[1] School of Informatics, University of Edinburgh, Edinburgh, United Kingdom
[2] UCB Celltech, Slough, Berkshire, United Kingdom

## ABSTRACT

Dynamic modelling has considerably improved our understanding of complex molecular mechanisms. Ordinary differential equations (ODEs) are the most detailed and popular approach to modelling the dynamics of molecular systems. However, their application in signalling networks, characterised by multi-state molecular complexes, can be prohibitive. Contemporary modelling methods, such as rule-based (RB) modelling, have addressed these issues. The advantages of RB modelling over ODEs have been presented and discussed in numerous reviews. In this study, we conduct a direct comparison of the time courses of a molecular system founded on the same reaction network but encoded in the two frameworks. To make such a comparison, a set of reactions that underlie an ODE model was manually encoded in the Kappa language, one of the RB implementations. A comparison of the models was performed at the level of model specification and dynamics, acquired through model simulations. In line with previous reports, we confirm that the Kappa model recapitulates the general dynamics of its ODE counterpart with minor differences. These occur when molecules have multiple sites binding the same interactor. Furthermore, activation of these molecules in the RB model is slower than in the ODE one. As reported for other molecular systems, we find that, also for the DARPP-32 reaction network, the RB representation offers a more expressive and flexible syntax that facilitates access to fine details of the model, easing model reuse.

In parallel with these analyses, we report a refactored model of the DARPP-32 interaction network that can serve as a canvas for the development of more complex dynamic models to study this important molecular system.

Corresponding author
Emilia M. Wysocka,
emilia.m.wysocka@gmail.com

## INTRODUCTION

Computational dynamic modelling probe mechanistic and quantitative aspects of molecular interactions, which can grant the development of mechanism-based therapies with more predictive power on outcomes of therapeutic interventions (*Xie et al., 2014*). Such interventions often target molecular signalling (*Li & Mansmann, 2014; Jia, Piña-*
*Crespo & Li, 2019*; *Volkow & Boyle, 2018*), characterised as a complex system of coupled interacting components, often proteins, forming networks that activity leads to non-additive effects (*Kitano, 2002*). Defining molecular reactions as a set of coupled ODEs has traditionally enabled dynamic modelling of molecular pathways (*Sible & Tyson, 2007*). ODE-based modelling is a powerful and acclaimed formalism with a long tradition. It has established standards, and various software tools that support and facilitate the formulation and analysis of ODE models (*Dräger et al., 2008*; *Hoops et al., 2006*; *Sible & Tyson, 2007*). However, the explicit enumeration of molecular species as variables required by the equation-based formulation excludes the representation of molecules that assemble into multivalent protein complexes; typically these have multiple functionally divergent states, a common characteristic of molecules involved in cell signalling (*Hlavacek et al., 2003*). Computational modelling methods effectively addressed these challenges of expressivity, increasing complexity of the modelled systems. An example of such a method is RB modelling, designed to model interacting proteins. Compared to the ODE-based paradigm, which represents the molecular system as concentrations of molecular species and focuses on their reaction kinetics, RB modelling is an agent-centred method in which the distribution of molecular compositions can be studied along with their abundance (*Faeder et al., 2003*; *Blinov et al., 2004*; *Danos & Laneve, 2004*).

## Previous comparisons

The potential of the RB paradigm has been extensively discussed (*Danos, 2007*; *Chylek et al., 2013*, *2014*, *2015*), exemplified (*Danos, 2007*) and used to answer novel biological questions across signalling, regulatory and metabolic networks (*Wilson-Kanamori et al., 2015*; *Antunes, Roque & Simoes-de Souza, 2016*; *Di Camillo et al., 2016*; *Santibáñez, Garrido & Martin, 2020*; *Chattaraj, Blinov & Loew, 2021*; *Nosbisch, Bear & Haugh, 2022*, see reviews for earlier models published before 2006—*Hlavacek et al., 2006*; and from 2007 to 2013—*Chylek et al., 2014*). These models are often based on chemical reaction networks previously developed for ODEs. Though the RB paradigm was implemented to match the concept of chemical reactions, rules are generators of reaction networks that may lead to diverging results. As the modelling paradigm can affect the underlying model specification, it would be informative to compare simulations of RB and ODE models defined by the same reaction network, originally built to be solved with ODEs. RB and ODE models have been compared on various aspects before. For instance, *Blinov et al. (2006)* compared network-like model of epidermal growth factor receptor (EGFR) based on reactions first defined in the ODE model of *Kholodenko et al. (1999)*, with simplified pathway-like structure. However, several assumptions underlying the original model were purposefully modified what changed the underlying interaction network, such as the dissociation of EGFR dimers when phosphorylated or bound to other molecules. In addition to contrasting rules to reactions, another intensively studied aspect, finalised with positive conclusions, was whether stochastic simulation can reveal new properties of a system previously modelled as deterministic one (*Vlysidis & Kaznessis, 2018*; *Hahl & Kremling, 2016*; *Bustos et al., 2018*). As simplistic as it may sound, none of the comparisons

mentioned were intended to painstakingly disassemble reaction using the rules of a relatively medium-sized model and compare simulation results.

## Objectives

This study compares the existing ODE model with the new RB model, where both are based on the same definition of chemical reactions. We specifically ask: (1) how closely can the dynamics of an ODE-based model be replicated with an RB one? (2) If differences between the two are observed, what is the underlying cause? (3) If the dynamics are indeed replicated, what advantages are there to using an RB model? We present the results of the models' comparison at the level of notation and model dynamics generated under different conditions. The advantages and disadvantages of the two model representations are discussed, alongside suggestions for future research. We hope that this small-scale attempt may nevertheless bring value to the modelling community to better choose between the two different approaches.

## BACKGROUND

Dynamic computational modelling frameworks consist of model specification and simulation methods. The model specification is a set of equations or instructions written in a machine-interpretable language that define relationships between variables. Models are run as numerical simulations using algorithms that calculate changes in the variables based on the model's formulation. By adopting a suitable level of abstraction and with the use of a sufficiently expressive language, systems can be modelled such that experimentally-derived evidence can be incorporated to improve the model's quality. A formal approach to the generation of models is desirable to (i) encode facts in an unambiguous and explicit manner, (ii) facilitate the understanding of models, (iii) allow easy modification of models to accommodate more than one hypothesis, (iv) aid interpretation of the underlying biological phenomena, and (v) provide a standard approach to the integration of novel data sources (*Kitano, 2002*). These features are especially important because model generation often requires knowledge spanning multiple disciplines; the existence of formal modelling frameworks enforces a common understanding of the explicit meaning of model components.

In the first step of building a kinetic model, transitions from reactants into products are defined as *chemical reactions* between molecules. Quantitative evaluation of model behaviour over time has been commonly achieved by converting coupled chemical reactions to a set of ODEs that are solved with numerical procedures (*Wilkinson, 2006*). Each rate equation expresses the change of concentration of a single molecular species over time, formulated with reaction rates that directly take part in the creation and elimination of this species (*Sible & Tyson, 2007*; *Klipp et al., 2005*; *Hlavacek et al., 2006*). Each reaction rate is weighted by a reaction-specific *rate constant*. Time courses obtained by solving ODE models are continuous and deterministic, characterised by smooth and gradual change of species concentrations over time (*Wilkinson, 2009*). Although this setup does not reflect the actual characteristics of subcellular events driven by random collisions between discrete molecules (*Gillespie, 1976*), this approach is correct as long as abundances of

reactants are large enough to render random fluctuations as negligible (*Chen et al., 2010*). If this condition is violated, the current development of machine coding formats for biological models (*e.g.*, SBML) allows deterministic solvers and stochastic simulators to be applied to the same model specification (*Gillespie, 1977*; *Hoops et al., 2006*). A more critical shortcoming of ODE-based models lies in the requirement for explicit enumeration of all molecular species in signalling networks (*Hlavacek et al., 2003*; *Danos, 2007*). This drawback precludes mechanistic modelling of systems with multi-state promiscuous molecules that can adopt combinatorially complex states (*Seshacharyulu et al., 2012*; *Chen et al., 2016*; *Mayer, Blinov & Loew, 2009*; *Suderman & Deeds, 2013*), where only a small fraction can be represented with ODEs (*Chylek et al., 2014*). However, the development of formal methods in computer science has expanded the number of observed properties of biological systems that can be dynamically and quantitatively modelled (*Bartocci & Lió, 2016*) (reviewed in *Machado et al., 2011*; *Tenazinha & Vinga, 2011*; *Bartocci & Lió, 2016*; *Ji et al., 2017*; *Le Novère, 2015*). As this study examines an alternative to ODEs, the focus lies on non-spatial single-scale mechanistic methods. Of those that fit this characterisation (*Baeten, 2005*; *Ciocchetta & Hillston, 2008*; *Regev, Silverman & Shapiro, 2001*; *Guerriero, Priami & Romanel, 2007*; *Dematté et al., 2010*) we chose rule-based modelling (RBM) as a suitable framework for representing systems with multistate combinatorial interactions. Among two major RBM implementations (*Stefan et al., 2014*), the very first being BioNetGen (BNG) (*Faeder, Blinov & Hlavacek, 2009*; *Faeder et al., 2003*; *Blinov et al., 2004*), the Kappa framework (*Danos & Laneve, 2004*) was chosen for this comparison. Although there are some notational differences between the two frameworks, the reactions are coded as rules in virtually the same way (*Suderman & Hlavacek, 2017*). An earlier study by *Suderman & Deeds (2013)* also showed equality of simulation results between both frameworks.

RBM is based on a formal concept of *graph rewriting*, where molecules are abstracted as structured graph objects referred to as *agents* with *sites* that express *internal* and *binding states*, together constituting *interface* of an agent (*Feret et al., 2009*). Reactions are defined as graph transformations encoded as *rules*, that are instructions for local and conditioned transformations. The reaction conditions are encoded in the agent interfaces. Depending on the completeness of these conditions, a rule can represent either a set of reactions that match the *pattern* expressed in a rule, or an exact reaction instance with complete information about the agents' states (*Feret et al., 2009*). Thus, a rule can express an infinite number of reactions with a small and finite number of generalised rules (*Chylek et al., 2013*). During the system simulation agents form *mixture* of *molecular species* that contain a complete description of their states and site occupancy, captured by *snapshots* that provide information on molecular species and their quantities at specified time-points. The trajectories resulting from the model simulation are obtained by declaring variables, called *observables*. If declared with an incomplete pattern, an observable is the sum of the trajectories of the many molecular species that fit the pattern. The Kappa language also allows to induce perturbations during the simulation, for example by updating rate constants or adding molecules. The Kappa simulator, KaSim (*Danos, 2007*; *Krivine, Danos & Benecke, 2009*), is based on Gillespie's Stochastic Simulation Algorithm (SSA) (*Gillespie,*

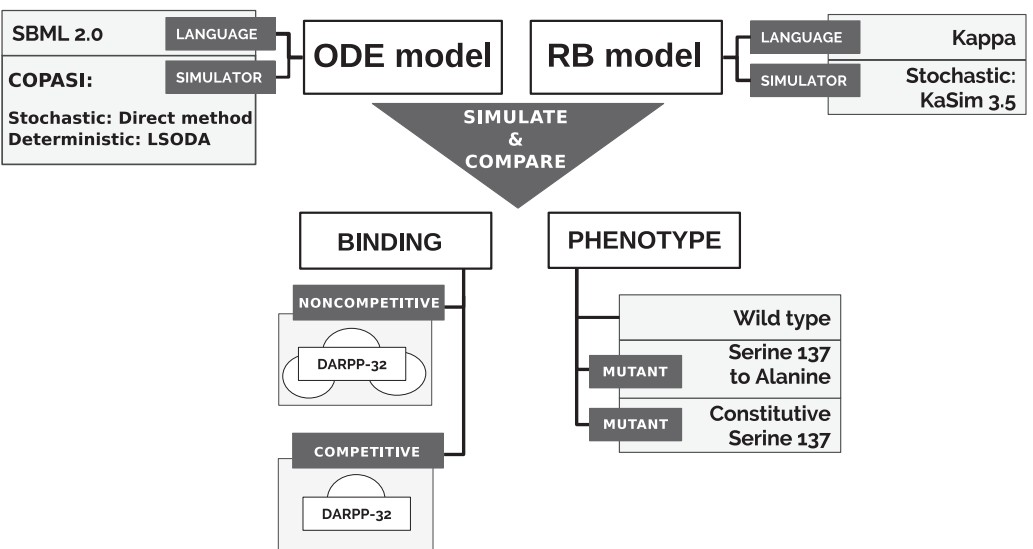

**Figure 1 Approach to comparison of ODE and RB modelling frameworks.**

*1976*, *1977*) that produces individual stochastic trajectories of molecules (*Gillespie, 1977*; *Wilkinson, 2009*), and whose performance is independent of the size of the reaction network (*Yang & Hlavacek, 2011*). This RB formulation gives a sufficiently expressive system to capture the principal mechanisms of signalling processes (*e.g.*, biding, dissociation, synthesis, degradation, and state change; *Liu & Thiagarajan, 2012*), providing site-specific details of molecular interactions (*e.g.*, affinities, dynamics of post-translational modifications, domain availability, competitive binding, causality) and the structure of interaction networks.

## METHODS

To directly compare models' simulations, we selected an ODE model available in a machine-readable format that can be numerically simulated in non-obsolete software, thus satisfying the reproducibility criterion. A model of the immediate interactors of dopamine- and cAMP-regulated neuronal phosphoprotein with molecular weight 32 kDa (DARPP-32) network by *Fernandez et al. (2006)* satisfies this requirement. The model was also considered as a solid core for the construction of larger and more complex models in the community interested in the modelling of dopamine (DA)-dependent synaptic plasticity (*Manninen et al., 2011*). Finally, it is a study widely cited not only by modellers (*Nair et al., 2016*; *Nakano et al., 2010*; *Mattioni & Le Novère, 2013*) but also by experimentalists (*Bales et al., 2011*; *Kim et al., 2015*; *Buesa et al., 2016*). It should be noted that the stochastic simulation for this particular system should not show any significant differences from the ODE model, as the number of particle copies is sufficiently large. The reactions underlying the ODE model of *Fernandez et al. (2006)* ("model B") were encoded into the Kappa language, version 3.5 (*Feret & Krivine, 2012*). Then, models were simulated in different variants to obtain time courses of equivalent observables to compare (Fig. 1). First, however, we briefly introduce the DARPP-32 interaction network.

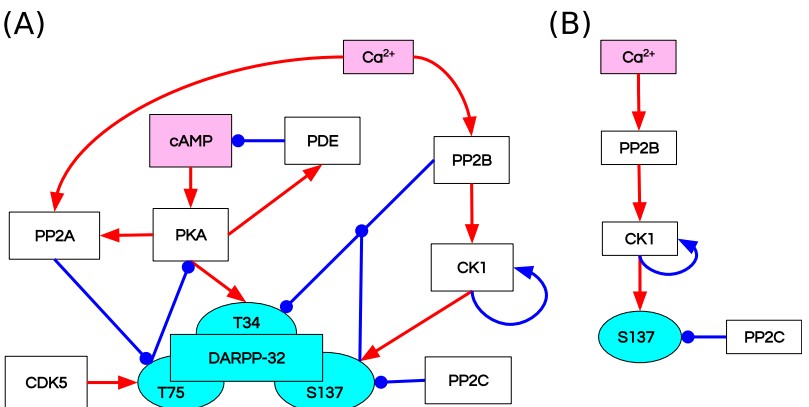

**Figure 2 Reaction diagrams showing (A) the DARPP-32 network included in the ODE model by** *Fernandez et al. (2006)* **Nodes: DARPP-32 (*cyan*), second messengers (*magenta*), kinases/ phosphatases (*white*).** Edges: inhibition reactions (*blue*), activation reactions (*red*); (B) the observables with the greatest divergence between trajectories of the ODE and RB models. These observables are connected in a chain of mutually dependent activation reactions triggered by the influx of calcium ions ($Ca^{2+}$).

## DARPP-32 and its interaction network

DARPP-32, Protein Phosphatase 1 Regulatory Inhibitor subunit 1B isoform 1 (PPP1R1B) (*NCBI Resource Coordinators, 2016*) is an important multistate and intrinsically disordered protein (*Marsh et al., 2010*) regulating DA-dependent synaptic plasticity in medium spiny projection neurons (MSPN), the predominant striatal cell type, where signalling cascades are activated simultaneously by glutamatergic and dopaminergic stimuli (*Beninger & Gerdjikov, 2005*). The malfunction of DARPP-32 relates to a number of neurological disorders, *e.g.*, Alzheimer's disease (*Cho et al., 2015*), addiction (*Philibin et al., 2011*), affective disorders (*Kunii et al., 2014*), and schizophrenia (*Kunii et al., 2014*; *Wang et al., 2015*). Glutamatergic and dopaminergic signals are integrated by DARPP-32 which is involved in a complex network of interactions regulated by its multiple phosphorylation sites, of which four are known to have an impact on DARPP-32 itself (*Yger & Girault, 2011*): Threonine 34 (Thr34), Threonine 75 (Thr75), Serine 137 (Ser137), and Serine 102 (Ser102). The Threonine sites (Thr34, Thr75) have major regulatory roles in signal processing. Whereas the Serine sites (Ser137, Ser102) reinforce Thr34 signal.

*Fernandez et al. (2006)* studied the integrative effect and sensitivities of DA- and glutamate (Glu)-mediated signals on the DARPP-32 network (Fig. 2A). Their model examined the particular effect of cyclic adenosine monophosphate (cAMP)-pulse followed by $Ca^{2+}$ spike trains showing that DARPP-32 is a robust integrator, far more complex than a bistable switch between DA and Glu signals. The authors included two main pathways that mediate these signals, cAMP–PKA–DARPP-32 phosphorylated at Threonine 34 (D34) and $Ca^{2+}$–PP2B–DARPP-32 phosphorylated at Threonine 75 (D75). Contrary to the majority of previous models of the Glu and DA signal integration (*Lindskog et al., 2006*; *Gutierrez-Arenas, Eriksson & Kotaleski, 2014*; *Nair et al., 2016*, *2014*), DARPP-32 included three phosphorylation sites: Thr34, Thr75 and Ser137. The authors performed two *in silico* mutagenesis experiments modifying the role of Ser137. The first mutation inhibits site

phosphorylation by changing Serine to Alanine at the 137 position (Ser137Ala). The second mutation leads to permanent phosphorylation of the Ser137 site (constSer137).

## Model translation

As a single reaction can be written as a single rule, the translation could have been accomplished in a one-to-one manner. However, to fully benefit from rule patterns, multiple reactions can be condensed into fewer rules by removing irrelevant context, *i.e.*, *decontextualised*. The context of a reaction in the RB model is defined as the information about the agent's binding sites, partners and internal states. Based on this definition of reaction context, the following criteria guided decisions about condensing reactions into rules. Given a set of reactions of the same type (forward, backward, or catalytic) between the same reactants (agents), if the difference between reactions lies in agent states (internal or binding) that do not change after the transition from reactants to products, and reaction constants (rates) in all these reactions have the same values, then information about agent states does not define reaction conditions; hence, it can be removed from the reaction notation, *i.e.*, a set of reactions become a rule pattern.

Among the least intuitive cases in encoding reactions into rules are complex substrate activations of PKA and PP2B. PKA is activated by the binding of four cAMP molecules, whereas PP2B activation requires four $Ca^{2+}$ ions. In other words, multiple molecules of the same type bind substrates on different sites that have to be uniquely named, which requires explicit encoding of all possible binding combinations on four different sides (*Danos, Koeppl & Wilson-Kanamori, 2011*), called hereafter as *combinatorial binding*. Translation from deterministic to stochastic rate constants and molecular concentrations to copy-numbers were performed in a standard way (*Feret & Krivine, 2012*; *Sekar & Faeder, 2012*). Lastly, the cAMP pulse and the $Ca^{2+}$ spiking are reproduced by the addition of molecular copy numbers and modification of rate constants during the simulation, respectively.

## Approach to comparison of models

The qualitative comparison of the results of two dynamic models required the simulation of both models in a stochastic scheme and the alignment of the trajectory of the corresponding observables under varying conditions to comprehensively compare the frameworks (Fig. 1).

### *Selecting and pairing observables*

The plots in the original publication show aggregated variables that are summed trajectories of multiple molecular species. For instance, "D34" denotes DARPP-32 phosphorylated at Thr34, regardless of its state of binding or other phosphorylation sites. The concept of aggregated variables corresponds to *observables* in RB modelling, and therefore, we use the term *observable* hereafter to denote aggregated variables. Observables of the ODE model were aggregated here based on their names matching partial strings representing the observables of interest. To verify this approach, obtained observables of the ODE model with this method were qualitatively compared with the six observables plotted in the original publication. The choice of other observables follows these principles:
(1) if an agent has internal states, the activated state is set as its observable form, *e.g.*, "CK1u"; (2) if an agent is created and degraded over the simulation, its observable is set to its least specific form, *e.g.*, "PKA"; (3) if an agent is not created and degraded during the simulation, *i.e.*, its level remains constant throughout the simulation and has no internal states, its observable is set to its bound form, *e.g.*, "_CDK5" (see Table S1 for the complete list of RB and ODE observables with definitions).

### Model simulation

We simulate both models in the stochastic scheme but within different simulation environments. The RB model was simulated with KaSim. The SBML format of the ODE model was run with COPASI (version 4.20), a common simulation environment for Systems Biology Markup Language (SBML)-formatted models (*Hoops et al., 2006*), using the deterministic solver (LSODA) and stochastic simulator (*direct method Gillespie, 1977*). To match units of molecular abundances across all simulation setups, the results of the SBML model were set to "particle numbers", equivalent to copy numbers in the RBM naming convention.

### Model perturbations

To induce the first type of site-directed mutations, Ser137Ala and constSer137, we modified rate constants during the simulation. In both cases, the alteration of the model involved the inactivation of four reactions by zeroing their rate constants. In the RB model, these reactions are represented by one rule, thus a change of a single constant induced each mutagenesis.

We additionally tested the RB model with two different binding schemes, applicable only to the RB model, and further called *noncompetitive* and *competitive binding* (Fig. 1). In the noncompetitive binding, all interactors of DARPP-32 can bind simultaneously to three different sites. The competitive binding assumes one interaction with DARPP-32 at a time, which reflects the ODE model assumption.

## RESULTS

Comparison of models was performed on two levels, model notation and simulation results. The model notation was analysed by dividing the model into components and comparing their sizes. We expect the set of reactions underlying the ODE model to be represented with fewer rules since a single one can constitute a pattern representing several reactions. The comparison of simulation results involved the alignment of equivalent time courses obtained by model simulations. We performed the comparison of time courses between three variants of each model: (1) base-line condition (wild-type) and two site-directed mutations: (2) Ser137Ala and (3) constSer137. Finally, we compared two RB model variants, representing: (1) DARPP-32 with a single binding site; and (2) DARPP-32 with three independent binding sites.

**Table 1 The specifications of the ODE and RB models can be broken down into elements, the number of which can be compared.**

| ODE model Model component | Total counts | Total counts | RB model Model component |
|---|---|---|---|
| Reaction instances | 152 | 132 | Reaction rules |
| Concentration-based rate constants | 152 | 62 | Stochastic rate constants |
| Initial concentrations | 75 | 8 | Initial copy numbers |
| Molecular species | 75 | 91/137 | Molecular species |
| Stimuli events | 21 | 21 | Stimuli events |

## Rule patterns reduce reaction number in a certain type of model components

Table 1 juxtaposes the total counts of model components in each model. The RB model represents 152 reactions with 132 rules, each parameterised by one of 62 unique rate constants. This number is lower than the total number of rate constants used to parameterise the ODE model (152). The final rule set is more than twice as large as the unique number of rate constants, meaning that more than one rule is parameterised by the same rate constant. The number of molecular species in the RB model, obtained with snapshots capturing the state of the molecular mixture over simulation time (every 10,000th event), is 91 for the competitive RB model, and 137 for the non-competitive one. In both cases, the sum of molecular species is higher than in the ODE model (75).

As expected, the number of rules corresponding to reactions is lower, and the number of molecular species is much higher, confirming that expression patterns reduce the number of rules needed to represent a reaction system. Nonetheless, the number of rules is only slightly lower than the number of reactions (152 to 131). If we closely compare models by parts representing more general molecular mechanisms, rule representation reduces the reaction number in some components but extends it in others (Table 2). The reduction occurred only in "DARPP-32 phosphorylation" and "PP2A activation by $Ca^{2+}$" components, where combinations of states of DARPP-32 phosphorylation sites do not have to be explicitly named. In contrast, the increase in the number of reactions compared to the reactions occurred in the components "PKA activation" and "PP2B activation". They both have four sites that bind the same molecules, $Ca^{2+}$ and cAMP, respectively.

## RB model recapitulates dynamics of ODE model with minor discrepancies

The RB model recapitulates the principal dynamics of the ODE model, albeit there are some observable differences (compare Figs. 3B and 3C). For instance, during the relaxation phase (after the 600th time point), "D34" in the ODE model needs 100 more time steps to reach the second peak, and it is weaker than its RB counterpart ("D34*"). Worth noting is that the standard deviation in the stochastically simulated ODE model reveals a distinctive variation in abundance of the "D34" observable during the relaxation phase. We further use the stochastic trajectories of the ODE model for comparison with the RB model. For a

**Table 2 The list of reactions in the *Fernandez et al. (2006)* publication was divided into components based on more general molecular processes represented by subsets of reactions, such as phosphorylation or activation.** We can closely examine the relationship between reaction and rule by comparing models by component. The table shows the number of reaction rules *vs* reaction instances and a unique number of rates per model component. It is noticeable that the reduction in the number of reaction instances due to the translation of reactions into Kappa language occurred in only two model components (1. & 8.), while in two others, it resulted in an expansion of the rule number (5. & 6.).

| | Model component | Reactions | Rules | Unique rate constants |
|---|---|---|---|---|
| 1. | DARPP-32 phosphorylation | 84 | 27 | 27 |
| 2. | CK1 phosphorylation | 4 | 4 | 4 |
| 3. | PDE phosphorylation | 4 | 4 | 4 |
| 4. | PP2A phosphorylation | 4 | 4 | 4 |
| 5. | PP2B activation | 4 | 24 | 4 |
| 6. | PKA activation | 12 | 56 | 7 |
| 7. | cAMP & $Ca^{2+}$ degradation | 8 | 8 | 8 |
| 8. | PP2A activation by $Ca^{2+}$ | 32 | 4 | 4 |

closer examination, traces of 15 observables (defined in Table S1) obtained from ODE and RB simulations were paired and superimposed (Fig. 4). Next to the clear matches (*e.g.*, Figs. 4B, 4E, 4H and 4N), there are discrepancies between paired curves. Five of these 15 observables (Figs. 4C, 4F, 4I, 4J and 4O) are examples of the largest divergence between models following a similar pattern of behaviour. They are directly connected in a chain of activation reactions that begins with $Ca^{2+}$ (Fig. 2B). Higher abundance of all $Ca^{2+}$ ions present in the system of the ODE model (Fig. 4C) could explain differences between the remaining four observables. However, the trajectory of "all_Ca" remains at the 0 level during steady states rising only in the spiking interval, which resembles the abundance of free $Ca^{2+}$ (Fig. 4B). $Ca^{2+}$ activates PP2B represented by the trajectory "PP2Bactive". The higher level of "PP2Bactive" is consistent with the other three observables, suggesting that this is a factor generating the divergences between the models. Based on the curves of the ODE model, we can reason that a stronger activation of PP2B results in proportionally more copies of the unphosphorylated CK1 and phosphorylated D137. This, in turn, increases substrate availability for PP2C; therefore, more copy-numbers of its bound form. This effect is inverted in the trajectories of the RB model.

## RB language allows for detailed dissection of observed molecular species

As the "all_Ca" observable trajectory produced by the ODE model is much lower than in the RB model at the steady state, and "PP2Bactive" appears to dictate the higher effect on the other three observables ("D137", "CK1u", "_PP2C"), "all_Ca" and "PP2Bactive" are closer analysed in further steps.

According to the reaction system underlying both models, the activated PP2B is a complex of four $Ca^{2+}$ ions and PP2B. This detail is not explicitly stated in the variable name of the ODE model. Therefore, to obtain the trajectory of all $Ca^{2+}$ ions, the sum of the

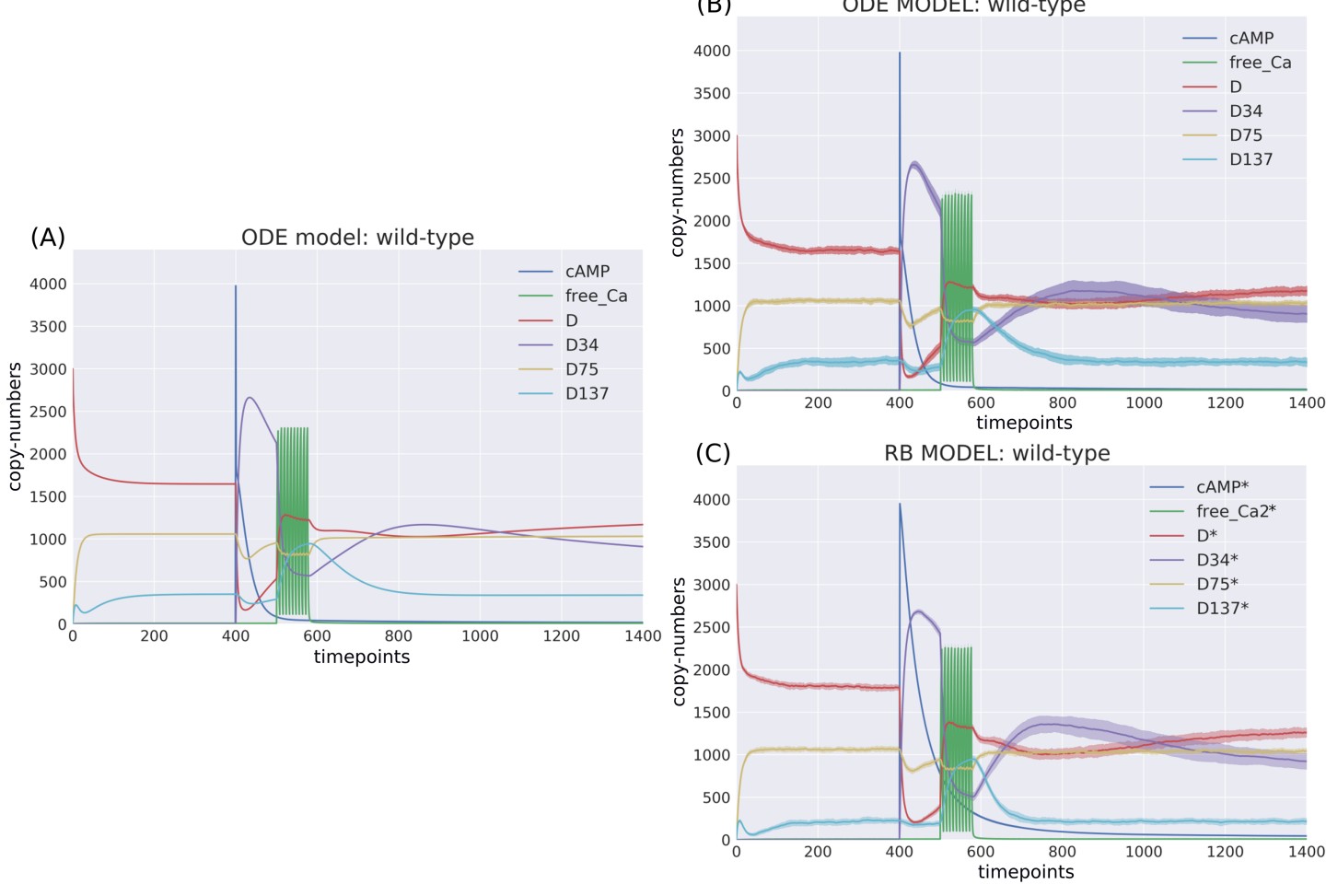

**Figure 3** Time-courses of the ODE model for DARPP-32 isoforms triggered by a pulse of cAMP followed by a train of Ca²⁺ spikes obtained with **(A) a deterministic solver, and (B) a stochastic simulation.** Trajectories of the stochastic simulation were obtained from calculating mean value (*line*) and standard deviation (*shade*) based on 40 simulations. (C) RB model (stochastic simulation). Variable isoforms of DARPP-32: "D"—unphosphorylated; "D137"—Ser137 phosphorylated; "D75"—Thr75 phosphorylated; "D34"—Thr34 phosphorylated.

copy numbers of the molecular species whose variable names contain "Ca²⁺", would have to be replaced by a more thorough analysis of the relevant reaction context of the ODE model. This is not the case in the RB model, where an observable of interest is obtained with an automated procedure that sums the trajectories of molecular species containing the specified expression pattern. Since the trajectory of all Ca²⁺ in the RB model includes the ions bound to PP2B, the comparison of "all_Ca" to "all_Ca*" is inaccurate due to a difference in the molecular species included in these observables. A similar inaccuracy, related to the naming of the observables, explains the discrepancy between the time courses of the total number of cAMP observables (Fig. 3). In contrast to the RB model, the multiple copies of cAMP bound to R2C2 are not included in this ODE model time course (Fig. 4A).

To determine the identity of molecular species whose trajectories were summed up in "all_Ca*", all species containing Ca²⁺ in the RB model simulation were isolated from snapshot data ammounting to 24 compared to only 13 in the ODE model ("all_Ca").

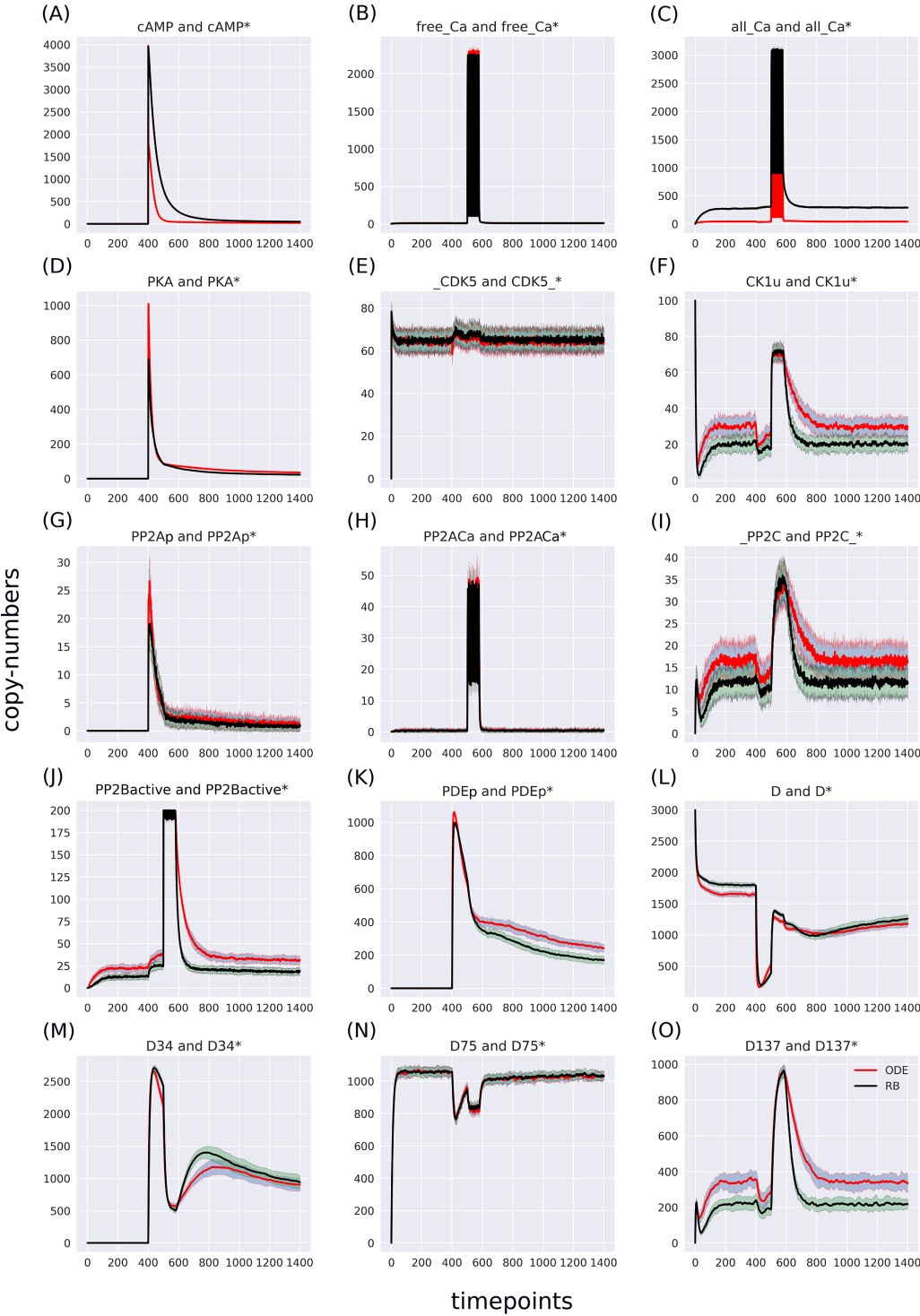

**Figure 4  (A–O) Superimposed time courses of stochastic variants of the ODE and RB models in the baseline condition.** Note that the scales on the y-axis are different to closely compare the traces of the observables. Trace colour: ODE (*red*), RB (*black*).

These 13 species correspond in molecular composition to 18 of the 24 RB species sampled in total. The six absent species in the "all_Ca" observable are composed of an active form of PP2B containing four $Ca^{2+}$ ions, either free or bound to phosphorylated CK1, or DARPP-32 in four different combinations of phosphorylation states. The number of species comprising "all_Ca" observable in the RB model is higher by five because the half-active form of PP2B (bound to two $Ca^{2+}$ ions) in the RB model exists in six variants. Whilst in the ODE model, it is represented as a single species, named "PP2BinactiveCa2". By rerunning the RB model with a new set of observables matching those in the ODE model and superimposing only one of six trajectories of "PP2BinactiveCa2", we can obtain a close match between the "all_Ca" observables (Fig. S1C). This demonstrates that the differences between the "all_Ca" observables of the two models can be explained by the difference in the number of representations of the molecular species. As the six trajectories have the same dynamics and average levels of abundances, choosing one of them is arbitrary (Fig. S2). Moreover, the distinction between locations of two $Ca^{2+}$ ions on the numbered sites of PP2B is irrelevant since all four sites are functionally indistinguishable.

## Rate constants of reactions formulated with "combinatorial-binding" notation should be increased to match ODE trajectories

The largest discrepancy between trajectories can be observed in "PP2BinactiveCa2" (Fig. 5M) that for the RB time course was obtained by summing six entities representing a half-active PP2B into one. If divided by six, representing a single variant of half-active PP2B, the trajectory of the RB model becomes lower than the one of the ODE model (Fig. S3M). These six forms of the half-active PP2B suggest that a better fit between the two models can be achieved by decreasing the constant rate of rules that represent the binding of $Ca^{2+}$ to free PP2B. However, the decrease of this rate constant of PP2B intermediate form would lead to a further decrease of copy numbers of other coupled observables that trajectories are lower than in the ODE model in the current parameter setting, *e.g.*, the fully active form of PP2B (Fig. 4J). To further examine this observation, we can return to the comparison of model specifications (Table 2). The four reactions of PP2B activation are represented by 24 rules, explicit in site-specific detail that includes all combinations of $Ca^{2+}$ positioning on four sites of PP2B. Moreover, instead of three distinct PP2B species in the reaction representation (inactive, half-active, activated), in the RB model there are eight different molecular species. Thus, the fine-grained representation of species in the molecular mixture slows down the transition from inactive to active PP2B, despite equal presence of the two-step transition encoded in both formalisms. The same can be observed in the second example that required a much larger number of rules, *i.e.*, an activation of PKA. Figure 4D shows that the RB trajectory of the "PKA*" observable also reaches a much lower peak than its ODE counterpart. Accordingly, values of rate constants of rules that the number increased due to the "combinatorial binding" notation in the RB model, *i.e.*, are represented by more variants of species, should be increased to closely match the ones in the ODE model.

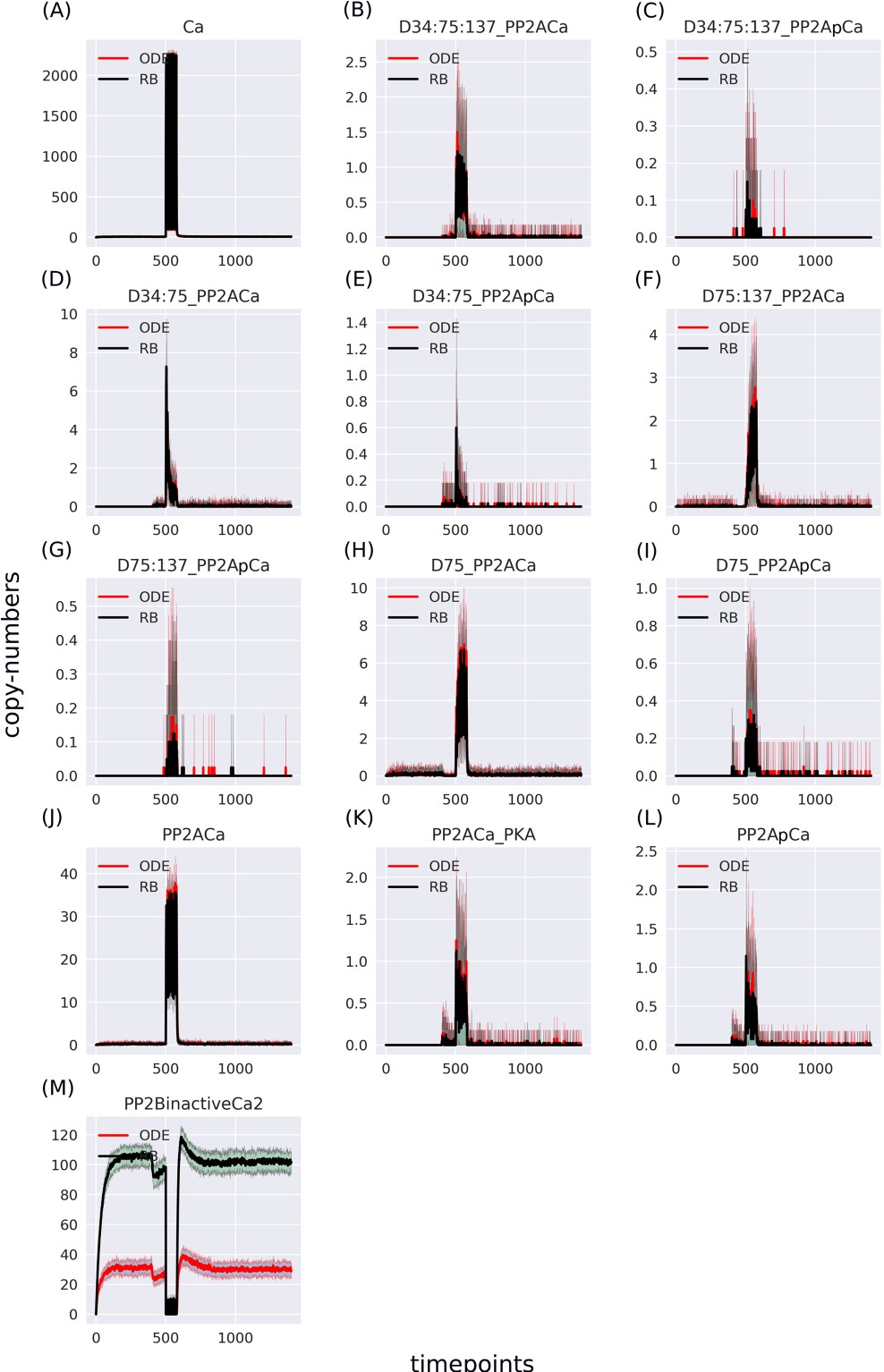

**Figure 5 (A-M) Traces of 13 pairs of molecular species containing Ca²⁺, selected to match the ODE model.** The largest disparity lies in the "PP2BinactiveCa2" variable—summation result of six entities representing an inactive form of PP2B in the RB model.

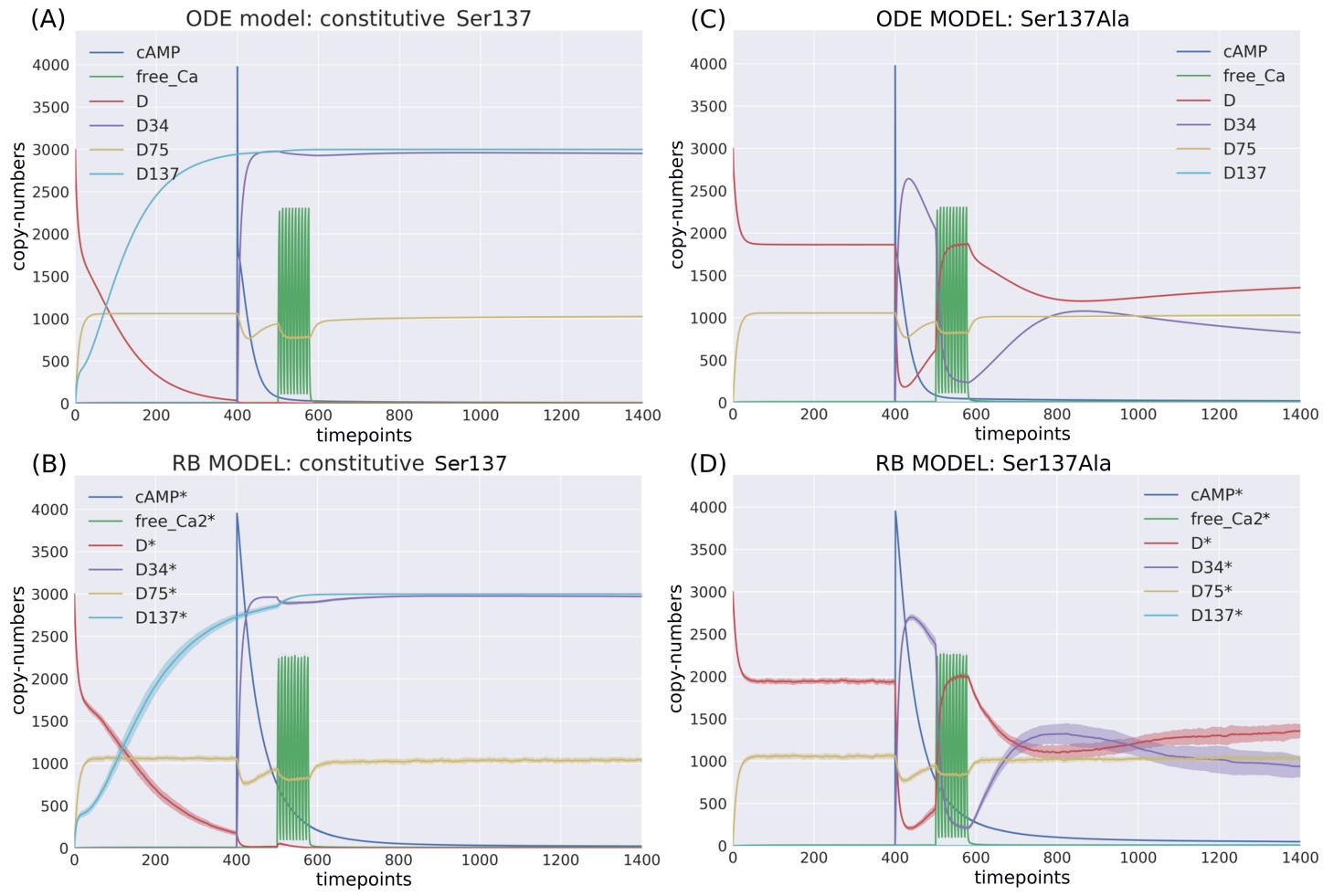

**Figure 6 Comparison of the constitutive Ser137 mutation induced in (A) ODE model in deterministic setting; (B) RB model in stochastic setting; and the Ser137Ala mutation in (C) ODE model in deterministic setting; (D) RB model in stochastic setting; The same interference performed on rate constants of the two models caused similar dynamics.**

## RB language facilitates modifications of dynamic models

### Site-directed mutations

The ODE model was analysed under two site-directed mutagenesis conditions affecting the Ser137 site. When these perturbations were applied in the RB model, we could observe a close fit in the initial conditions and a general pattern of dynamics based on six key observables (Fig. 6). Thus, RBM allows to emulate experimentally observed perturbations similarly to ODEs.

### Competitive and non-competitive site binding

We also compared two variants of the RB model with different binding site specifications to establish whether the dynamics of the model are affected when DARPP-32s binds multiple partners at once. The first specification is a competitive variant of the model with one binding site (oBS). In the second, a non-competitive variant, the partners bind simultaneously (three-binding-sites, tBS). This type of comparison between ODEs and RBs

was previously presented by *Blinov et al. (2006)*. In contrast to their results, the superimposed trajectories of two models (Fig. S4) demonstrate no effect on model dynamics. As the direct consequence of this modification is an increase in the size of the complexes to more than two proteins, it seems that larger complexes are rarely formed during the simulation. This interpretation is confirmed in a direct examination of species counts (Fig. S5).

## DISCUSSION

ODE-based modelling is a classical and commonly used method for creating detailed dynamic models of biological systems (*Lotka, 1920*; *Hodgkin & Huxley, 1952*; *Lisman, 1989*). It is frequently a point of reference and comparison to newly proposed modelling methods (*Morris et al., 2010*; *Ciocchetta & Hillston, 2008*; *Chaouiya, 2007*; *Chylek et al., 2015*; *Danos & Laneve, 2004*). Nevertheless, modelling of signalling systems with ODE poses difficulties due to complexities underlying molecular interactions (*Kholodenko, 2006*; *Stefan et al., 2014*). RBM was proposed as a solution to this problem.

Numerous reviews (*Danos, 2007*; *Chylek et al., 2014*; *Hlavacek et al., 2006*) and studies discussed the advantages of RB modelling over ODE. This article presents results of encoding reactions underlying an ODE model to the RB language and a comparison of their specification and simulation results. The manual translation of this ODE model into any RB language was necessary despite the existence of a method (*Tapia & Faeder, 2013*) for automated translation of the SBML-format encoding ODE-based models to an RB model format. The translation failed to generate executable model with correctly identified agents (Supplement 1).

### Effects of the RB framework on the model notation

Encoding reactions into rules slightly reduced the size of model specification and increased counts of molecular species, which confirmed the well-known advantage of rule representation (*Hlavacek et al., 2006*; *Chylek et al., 2014*). Closer analysis of reaction subsets representing more general molecular mechanisms showed that reduction in the reaction number is only true for reactions occurring between the same reactants, describe the same transformation, and are parameterised with the same values of rate constants, but differ only concerning binding or internal state of reactants. In this type of reaction, the number of unique reaction rates was equal to the number of rules. An increase in the number of rules representing reactions occurred where the same partner binds an agent at multiple sites. In such reactions, all possible positions and stages of the binding process had to be explicitly encoded. The "combinatorial binding" notation is not a general property of the RB language but characteristic of the Kappa syntax. In the alternative to Kappa RB framework, BNG, a rule is definable with identical sites' names. This implies that the rule pattern defined for one applies to the others, effectively shortening the rule description (*Sekar & Faeder, 2012*).

## Discrepancies in dynamics between models

As in other similar study (*Blinov et al., 2006*), the comparison of trajectories showed agreement between model dynamics, with some discrepancies. Their source is not due to the issue of ambiguous molecularity that can arise in Kappa, which requires control on the part of the modeller, as none of the rules in the DARPP-32 model allow the formation of closed rings between agents. The discrepancies mainly appeared due to a lack of precision in the variable names of the ODE model caused by differences in molecular species comprising tracked observables. When the simulation of the RB model was performed with observables exactly matching the ones in the ODE model, almost all paired trajectories fitted perfectly. The only problematic observable involved the "combinatorial binding" notation. Further analysis suggested that activation of proteins encoded with fine detail was slower than in the ODE model. More specifically, although half-active PP2B was more abundant in the RB model, its active form had lower levels compared to the ODE model. This observation may apply to RBM in general because the molecular mixture when simulated with NFsim (*Sneddon, Faeder & Emonet, 2011*), a variant of the network-free simulator for BNGL, will implicitly contain the same number of molecular species with different binding variants as it is in KaSim. Further, based on the example of antibodies with identical antigen-binding sites, the BNGL developers demonstrated that the equivalent set of rules in both RBM frameworks are parameterised by the same rate constant (*Suderman & Hlavacek, 2017*). In the BNG framework, however, this rate is automatically scaled by a factor equivalent to the multiplicity of indistinguishable ways to obtain the reaction product (*Faeder et al., 2005*; *Faeder, Blinov & Hlavacek, 2009*). Moreover, comparison of the time traces of Kappa- and BNGL-defined rules shows equivalent results under stochastic realisations (*Suderman & Hlavacek, 2017*). Thus, we argue that "combinatorial binding" expressed as a single rule in BNGL will not change the dynamics observed with Kappa. These specific discrepancies should be taken into account when the reactions and rate constants of RB models are derived from ODE models, and preferably determined experimentally for these reaction types.

## A shift in modelling focus with the use of the RB framework

The process of encoding reactions into rules turns attention to questions such as how many binding partners can simultaneously bind a protein. The translation process has shown that information about interfaces of interacting proteins and their alternative states would considerably ease the process of model development by guiding decisions on agents' signatures. Therefore, data resources that could support RB modelling, such as a source of protein interaction interfaces, post-translational modifications (PTMs), and protein domains. For instance, proteins containing phosphatase catalytic domains are enzymes of dephosphorylation reactions (*Sacco et al., 2012*). However, such detailed information is not accessible for most molecular agents.

## Facilitation of modification and reuse of dynamic models

Molecular species defined in the ODE framework are fixed and definite, whereas, in the RB model, they are a subject of investigation. The determination of created species and their

abundances is enabled with snapshots, state vectors of molecular mixture sampled during the simulation. Snapshots have been used as a tool for model exploration in other studies (*Sorokina, Sorokin & Armstrong, 2011*; *Suderman & Deeds, 2013*), and their visualisation is included as a standard tool in the Kappa framework (*Boutillier et al., 2018*). Although, unlike Kappa, BNG allows counting the number of species through network generation, this option is limited by the size of the model. Then, if not sampled, the number of specie types is calculated analytically (*Suderman & Deeds, 2013*). While this is an approximate approach, by using snapshots, we were able to study the emerging molecular species in detail and precisely track the observables of interest. Furthermore, the automatic identification and merging of time courses into observables makes the RB framework particularly advantageous over the ODE framework, as it avoids errors in the identification of molecules in the modelled system. This is important when the system consists of many molecules and their states are to be analysed in detail. In ODE models, on the other hand, obtaining the details of the molecule composition, hidden in the arbitrary names of the individual molecular species, would require further deconstruction of the reaction system. Though, the precise identification of molecules within species could be performed by parsing the SBML-model encoding, the *Fernandez et al. (2006)* model web page in the BioModels website (http://www.ebi.ac.uk/biomodels-main/BIOMD0000000153), shows incomplete annotations of molecular species behind variable names, both concerning the actual counts of interactions (*e.g.*, $Ca^{2+}$) and their components.

The *Fernandez et al. (2006)* study does not discuss if DARPP-32 partners bind to the same or different active sites, though DARPP-32 is an intrinsically disordered protein with an unknown binding interface (*Dancheck, Nairn & Peti, 2008*; *Mollica et al., 2016*; *Engmann et al., 2015*; *Choy, Page & Peti, 2012*). The ODE model specification demonstrates that DARPP-32 forms at most heterodimers. This type of modification in the ODE model would require the arduous extension to much more complex model by enumeration of additional molecular species, the addition of new equations and updating the existing ones. Contrary to this, a definition of such a binding scenario in the RB notation requires the same number of rules, provided that concurrently bound interactors do not influence each other. The RB model was tested with two types of site-directed modifications, demonstrating the framework's flexibility to reproduce experimentally conducted perturbations. Though the binding site modification effectively changed the model reaction network, it did not affect the model response. Nevertheless, this intervention demonstrated the ease of performing such alterations within the RB framework. Additionally, the pattern notation improves model clarity and provides an intuitive representation of a model akin to a set of chemical reactions rather than equations, potentially improving the learning curve for a modeller-to-be.

## Note on simulation time

The ODE simulation remains incomparably faster than the RB simulation. In both deterministic and stochastic settings, COPASI returns ODE results in seconds, while KaSim 3.5 requires ~40 min[1] to return the RB model results (single core on ThinkPad Core i7-4700MQ; RAM: 16GB). With more recent versions of KaSim (≥4), the execution

---

[1] The total CPU time measured with the "time" command on Linux OS.

time could be much shorter thanks to a new solution for the update of observable instances (*Boutillier, Ehrhard & Krivine, 2017*). Nevertheless, with a sufficiently simple model, network-based simulation can be performed far more efficiently using BNGL, bearing in mind that more complex models will have a higher runtime (*Sneddon, Faeder & Emonet, 2011*). When generating large-scale models, the type of RB model simulation can be scheduled on a case-by-case basis (*Santibáñez, Garrido & Martin, 2020*).

## Further explorations of the DARPP-32 RB model

There are two main routes for further explore the RB model. The first one is a modification of parameters defining different phases of combinatorially bound $Ca^{2+}$ ions to PP2B and cAMP to R2C2. A particular task would be to identify factors by which the binding constants could be modified to counteract the many intermediate variants of these complexes and the lower copies of their activated final forms. Next, it would be interesting to identify conditions under which we could observe a difference in model dynamics after the addition of binding sites to DARPP-32. In the current setup, the lack of difference might be caused by the similarity in occupancy between a single site and all three sites together, as they do not counter each other's binding properties. The probability of a site being connected depends on copy numbers of reactants and the strength of binding affinities. Reactions in the model are classified as weak, with dissociation rates in the range of $\mu M$. Low-affinity bindings generally lead to lower levels of site occupancy. Moreover, the amount of DARPP-32 molecules exceeds the total counts of all its interactors. Thus, with the current proportions of reactants, all three DARPP-32 sites cannot be saturated. To expose the potential differences in dynamics between the two binding scenarios, the site occupancy could be modified by increasing the size of the reactant pools or reducing the level of DARPP-32. As mentioned by *Fernandez et al. (2006)*, levels of DARPP-32 vary considerably, between $\mu M$ to tens of $\mu Ms$, in the striatum. With the greater availability of single-cell techniques for protein quantification (*Lo et al., 2015*) it would be worth establishing more precisely the range of DARPP-32 even at the resolution of a dendritic spine (*Otmakhov & Lisman, 2012*). Estimating variability between cells could also be used to compare the varying levels of phosphorylated DARPP-32 at Thr34 that were observed in the stochastic simulations.

## Extending the model to match evolving knowledge on the DARPP-32 interaction network

The main advantage of RBM over ODE is the flexibility to extend and manipulate the model, rather than to rewrite it into a new one. Through a single representation, the community interested in DARPP-32-related molecular systems could pool their knowledge. We believe that the refactored model of the DARPP-32 in a rule-based framework will facilitate the creation of such an updated model in future. It should be noted that the "DARPP-32 events" pathway (R-HSA-180024.3) specified in the REACTOME database is in fact the model of *Fernandez et al. (2006)* used in this study. So far, only early signalling events of DARPP-32 have been modelled, localised mainly in the cytosol. The Ser102 site regulates nuclear transportation of DARPP-32 representing

late signalling events (*Stipanovich et al., 2008*). The site was omitted in the *Fernandez et al. (2006)* model due to the lack of evidence that it may be affected by DA or Glu signalling (*Girault et al., 1989*; *Svenningsson et al., 2004*). However, a recent study by *Nishi et al. (2017)* suggests that Glu can decrease the effect of DA signalling by dephosphorylating DARPP-32 at Ser102 causing the accumulation of DARPP-32 in the nucleus, the effect known to be promoted by drugs of abuse (*Stipanovich et al., 2008*). Moreover, the role of the DARPP-32 interaction network in the nucleus appears to be an important future goal of the recently published ODE model incorporating DARPP-32 (*Yapo et al., 2018*), indicating the need to explicitly demonstrate the contribution of DARPP-32 to the observed switch-like behaviour of PKA in the nucleus. The inclusion of the Ser102 site seems inevitable to achieve this goal, but extending the current model with two phosphorylation sites may prove to be quite complex. Encouragingly, to calibrate this extended model, measurements experimentally obtained by *Nishi et al. (2017)* for all four phosphorylation sites could be fed into Pleione (*Santibáñez, Garrido & Martin, 2019*), a recently developed tool for this task in RBM. Lastly, it would be worth to establish if stochastic simulation can demonstrate new behaviour of this particular molecular system that deterministic simulation has so far not revealed.

### Need for formal prioritisation methods of emerging molecular species

RB modelling offers tools for the dissection of emerging molecular species during simulations. As the number of such molecular species increases it becomes difficult to identify which model components are of particular importance to the system. There is great potential to exploit commonly used methods for model exploration, such as sensitivity analysis, to identify critical model features including parameters and model output variables. It would be advantageous to support the modeller's assumptions with automated methods to prioritise model outputs for downstream analysis and to gain greater insight into the underlying biological systems.

## CONCLUSIONS

Dynamic molecular modelling has become increasingly important in uncovering and integrating our dispersed knowledge of molecular mechanisms. Choosing the best formalism to meet this challenge is a difficult task. In this work, we have presented a detailed and systematic comparison of two major formal approaches to quantitative modelling. We demonstrated the advantages and disadvantages of RB modelling to the prominent ODE approach. We confirm, after other similar studies, that RB modelling is a more detailed and flexible way to represent biological molecular systems, enabling exploration of individual molecular entities, model extension, and future reuse. These conclusions confirm the potential of the RB formalism and hopefully will embolden future exploration and research in this topic.

## LIST OF ACRONYMS

| | |
|---|---|
| **BNG** | BioNetGen |
| **$Ca^{2+}$** | Calcium ions |

| | |
|---|---|
| **cAMP** | cyclic adenosine monophosphate |
| **DA** | dopamine |
| **DARPP-32** | dopamine- and cAMP-regulated neuronal phosphoprotein with molecular weight 32 kDa |
| **EGFR** | epidermal growth factor receptor |
| **Glu** | glutamate |
| **MSPN** | medium spiny projection neurons |
| **ODE** | ordinary differential equation |
| **PTM** | post-translational modification |
| **RB** | rule-based |
| **RBM** | rule-based modelling |
| **SBML** | Systems Biology Markup Language |
| **Ser102** | Serine 102 |
| **Ser137** | Serine 137 |
| **SSA** | Stochastic Simulation Algorithm |
| **Thr34** | Threonine 34 |
| **Thr75** | Threonine 75 |

## ACKNOWLEDGEMENTS

The authors would like to thank Andrew Miller and Evangelia Petsalaki whose suggestions have contributed to the improvement of this manuscript.

### Funding

The project was jointly founded by UCB PHARMA S.A., acting through its UK subsidiary UCB CELLTECH, and the University of Edinburgh. The funders had no role in study design, data collection and analysis, decision to publish, or preparation of the manuscript.

### Grant Disclosures

The following grant information was disclosed by the authors:
UCB PHARMA S.A, UK subsidiary UCB CELLTECH.
The University of Edinburgh.

### Competing Interests

Matthew Page and James Snowden are employees of UCB Celltech.

### Author Contributions

- Emilia M. Wysocka conceived and designed the experiments, performed the experiments, analyzed the data, prepared figures and/or tables, authored or reviewed drafts of the article, and approved the final draft.

- Matthew Page analyzed the data, authored or reviewed drafts of the article, and approved the final draft.
- James Snowden analyzed the data, authored or reviewed drafts of the article, and approved the final draft.
- T. Ian Simpson conceived and designed the experiments, analyzed the data, authored or reviewed drafts of the article, and approved the final draft.

## Data Availability

Models and code to reproduce published figures are available at GitHub: https://github.com/ewysocka/rb_vs_ode_model_of_darpp-32.

## Supplemental Information

Supplemental information for this article can be found online at http://dx.doi.org/10.7717/peerj.14516#supplemental-information.

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
