# Peer review of "Comparison of rule- and ordinary differential equation-based dynamic model of DARPP-32 signalling network"

_PeerJ, doi:10.7717/peerj.14516_

## Round 0.1 · original submission · Major Revisions

We have received a series of critical remarks. The manuscript needs revision. Please check the references mentioned by reviewer #2. We may add a third review later. However, I think there are enough comments for the revision to be fixed.

Reviewer 1 ·

Basic reporting

Wysocka et al. developed a Rule-Based Model (RBM) for the DARPP-32 signalling network based on a published, ordinary differential equations (ODE) based model to answer if an RBM can recapitulate the predicted dynamics of an ODE model. Consequently, if an RBM can show further mechanistic details about activation of DARPP-32 that an ODE model cannot, thus showing that an RBM compared to ODE model has advantages. The authors presented the results of simulations, from which they were able to conclude that a rule-based model is a suitable and better choice to study the DARPP-32 signaling pathway than an ODE model. In addition, the authors included supplementary information in a well-structured online repository.

Experimental design

I would suggest to the authors to change the focus of the study and present an updated computational model for DARPP-32 written in kappa language or BNGL. Once developed, the authors can compare the predicted stochastic dynamics to the deterministic simulations and more importantly, to experimental data as the purpose of a model is to resemble quantitatively to experimental data and not to previous modeling attempts. It would even better if the authors use pySB (https://www.embopress.org/doi/full/10.1038/msb.2013.1) to encode the model and harness the pySB export/simulation capabilities to obtain the stochastic and deterministic simulations. Therefore, the research question seems not relevant and neither rigorously investigated, and in my opinion, the methodology is insufficient to replicate the modeling in the kappa language. Use of updated software is encouraged (e.g. KaSim 4.1), but not necessary. Also, the authors should consider using BNGL to encode “combinatorial binding”, as identical sites in the agent definition do not require to be named differently, and the authors should use and inform an elevated number of simulations (100-1000 or more even). It is encouraged to use numerical methods to compare dynamics, not “visual comparisons” (line 234), and convert the kappa model into ODE to determine the exact number of observable (i.e., do not use snapshots).

Validity of the findings

The author compared extensively both modeling formalism (for the signaling pathway), however, the most important issue is it is hard to agree that “no direct comparison of the time courses […] has been made before”. For instance, Santibanez et al, 2020 (Figures 2C-2E, https://academic.oup.com/bioinformatics/article/36/22-23/5473/6050699) showed that the predicted dynamics can differ when a deterministic simulation is compared to stochastic simulations, or no comparison can be performed due to excessive combinatorial explosion. Similarly, Bustos et al, 2018 (https://link.springer.com/protocol/10.1007/978-1-4939-8618-7_1) presented a compartmentalized Lotka-Volterra rule-based model that predicted extinction, contrary to the classical ODE model. Similarly, stochastic simulation has been compared to deterministic simulations (e.g., https://www.mdpi.com/1099-4300/20/9/678 and https://www.frontiersin.org/articles/10.3389/fgene.2016.00157/full). Even if we agree that the comparison (ODE vs RBM) is the first of its kind, the manuscript lacks quantitative comparison between dynamics (comparisons are visual), comparing one model written in two formalisms cannot be generalized, the RBM in kappa language can be fundamentally different from the ODE model as some sentences throughout the manuscript appear to describe (line 228; lines 372-393), and the model can simulate unintended transformations. Regarding the latter, the kappa language can encode in the same rule unimolecular and bimolecular reactions that can occur at the same or different rate (See Kappa Manual, section 3.7 “Ambiguous molecularity” https://kappalanguage.org/sites/kappalanguage.org/files/inline-files/Kappa_Manual.pdf). In contrast, BNGL can encode without ambiguity unimolecular and bimolecular reactions employing its dot-plus notation. Finally, as the authors are aware, a one-to-one conversion of an ODE model can be done using Atomizer and the GitHub repository shows successful attempts to convert the ODE model made back in 2017 (in contrast to the authors’ claim in lines 432-433), but the authors decided to develop the model using the kappa language without explanation of their decision. I was able to convert the ODE model into BNGL using the latest Atomizer software (from 2018).

Additional comments

As a minor and final point, in my opinion, there is an excessive amount of figures and sub-figures (12 figures in the manuscript plus 4 in the supplementary material) and unnecessary sections (“Modelling signalling systems with ordinary differential equations”, “Rule-based modelling - formal computational method for simulating molecular interactions”, and “Representative ODE model”) that can be reduced or referenced to already cited works.

Reviewer 2 ·

Basic reporting

- Text is well presented.
- Logical structure.
- Figures are presentable.
- The raw data is inadequate. The GitHub repository contains multiple folders with many different models, it is not clear where the master model that corresponds to the original SBML code is.
- Acceptable read-me materials needed at the GitHub repository, so a user can independently install kappa and follow examples to reproduce figures.

References:
The authors are perhaps unfamiliar with biological applications of rule-based modeling. For “models answering new biological questions” the authors cite the following references do not answer any biological questions:
- Kohler et al., 2014 - conference proceedings is not indexed in biological databases
- Chylek et al., 2014 and Hlavacek et al., 2006 – these publications are just reviews.
The only rule-based model answering any biological questions the authors cite is Antunes et al., 2016 – but it is implemented in BioNetGen, not Kappa.
If the authors would like to justify the value of rule-based modeling in biology, they should cite multiple publications in high-profile biological journals that use rule-based modeling. I can suggest several such models implemented with BioNetGen. To name few - JI 2003 https://pubmed.ncbi.nlm.nih.gov/12646643/ , JI 2012 https://pubmed.ncbi.nlm.nih.gov/22711887/, Nat Methods 2012 https://pubmed.ncbi.nlm.nih.gov/31363225/ , ELife 2021 https://pubmed.ncbi.nlm.nih.gov/34236318/, JBC 2022 https://pubmed.ncbi.nlm.nih.gov/35367415/.
While the authors mention BioNetGen, the other popular framework, they do not cite it, while the first version of this software was used in biological modeling in 2003 and described in 2004 https://pubmed.ncbi.nlm.nih.gov/15217809 and the current version in 2009 https://pubmed.ncbi.nlm.nih.gov/19399430/. NFSim (https://pubmed.ncbi.nlm.nih.gov/21186362/) which is a direct analogue of Kappa stochastic simulator is also never mentioned.

Experimental design

The manuscript provides a useful introduction to rule-based modeling. It describes it by taking an existing ODE model and converting it into a rule-based model using Kappa framework, one of the rule-based frameworks. However, the statement that “no direct comparison of the time courses of a molecular system encoded in the two frameworks has been made before” is clearly wrong. An identical approach of taking an existing ODE model (Kholodenko et al., 1999) and converting it to a rule-based model was implemented in 2006 in https://pubmed.ncbi.nlm.nih.gov/16233948/. The answer the authors of the manuscript provide “Kappa model recapitulated the general dynamics of its ODE counterpart with minor differences” is obvious by design, as a rule-based model is designed to match the original model described by a reaction network. The authors of a 2006 manuscript also discussed biological significance of the model conversion (e.g. effect of multiple sites), something completely omitted here. Other statements in the publication are also trivial:
- “These differences occur whenever molecules have multiple sites binding the same interactor” –multiple sites introduce new reactions that were not present before.
- “The notation of such rules requires a complete listing of all possible binding configurations.” – this is Kappa-specific issue, not BioNetGen
- “Furthermore, activation of these molecules in the RB model is slower than in the ODE one but can be corrected by revision of the rate constants used in the relevant rules”. – This issue was briefly discussed in https://pubmed.ncbi.nlm.nih.gov/16233948/
- “We conclude that the RB representation offers a more expressive and flexible syntax that eases access to fine-grain details of the model, facilitating model reuse” – all the cited reviews talk just about it.
Sections of the manuscript “Rate constants of reactions formulated with “combinatorial-binding” notation should be increased to match ODE trajectories”, and “Competitive and non-competitive site binding” were discussed in https://pubmed.ncbi.nlm.nih.gov/16233948/ and then in https://pubmed.ncbi.nlm.nih.gov/19399430/ .
Finally, in discussion, “It takes almost 40 minutes to simulate this particular RB model with the KaSim simulator. The solution of the ODE model in the COPASI environment in the deterministic setting returns in an instance.” – using BioNetGen for network generation would lead to simulation time of seconds to minutes. Also it would be instructive to simulate the model stochastically with NFSim.

Validity of the findings

This manuscript findings are similar to a research that recodes a very simple Java program to C++ and try to make any conclusions based on a simple single example.

Reviewer 3 ·

Basic reporting

.

Experimental design

.

Validity of the findings

.

Additional comments

This submission compares the behavior of a rule-based model and an ODE model.

This is interesting, yet, I find the message quite confusing.
As far as I understand, the differences that are reported come more from differences in modeling details, rather than on the execution paradigm.
I acknowledge that a paradigm can favor less/more detailed modeling practice since providing details may quiclky become combersome when the paradigm does not offer the right level of abstraction. But in this case, it is more reaction-based VS rule-based than ODE VS rule-based. But nevertheles, it is a modeling choice to fix which amounts of details shall be put in the model.
More precisely, I do not really understand why the authors have not used ODE rule based-models. It would have been more insightful I think to compare the stochastic execution and the ODE one of the same Kappa model, then the stochastic execution of the Kappa model with the SBML one, then the ODE execution of the Kappa model with the SBML one.

(Note that KaDE can be used to compile rule-based models into ODEs)
Ferdinanda Camporesi, Jérôme Feret, & Kim Quyên Lý. KaDE: a Tool to Compile Kappa Rules into (Reduced) ODE Models. In Computational Methods in Systems Biology, tools paper track (CMSB 2017), In: Lecture Notes in Computer Sciences / Lecture Notes in BioInformatics, volume 10545. © 2017, Springer.
http://dev.executableknowledge.org/docs/KaSim-manual-master/KaSim_manual.htm
https://www.di.ens.fr/~feret/CMSB2017-tool-paper/


In fine, the reported discrepancies are quite expected. There is no reason why competetive and non competitive bindings should have the same dynamics. This is the same for global phenomena that are modeled with a different number of intermediary steps (It would also have been important to recall that using several intermediary steps do encode a global phenomena is usefull to tune the time-probability distribution of a mechanism (to go beyond the exponential time-distribution).

---

## Round 0.2 · Minor Revisions

Thanks for the manuscript update and answer to the reviewers. There are some comments demanding revision. Please pay attention to the presentation style as suggested by reviewer #1.

Reviewer 1 ·

Basic reporting

I have no further comments. The authors addressed my concerns in their manuscript.

Experimental design

The authors should explain how did they convert from concentration into "copy-number", as the relevant ODE-based simulations show "copy-number" in their y-axes (Figures 3A, 3B, 4, 5, 7, 8A and 8C), not concentration. Otherwise, I have no further comments regarding the experimental design.

Validity of the findings

I have no further comments regarding the validity of the findings. The authors discussed properly the limitations of the approach to compare traces between an ODE-based model and an RBM.

Additional comments

In my opinion, the manuscript still shows unnecessary information, and it is structured in a way that can be hard to grasp or it can intimidate the reader (it is a 17-pages long manuscript with 8 figures and 2 tables). For instance, the Introduction section is a unique paragraph (or it looks like a single paragraph since line 70 ends exactly at the right border) that will be more readable if it is split into smaller sections. The Background section contains the absolutely necessary "DARPP-32 and its interaction network" subsection, but in my opinion, it should be under the Methods section, while referencing the remaining information during the introduction or including it in the Supplemental material. The Figure 1 and 5 might be combined into a unique figure.

Also, the authors should look at minor style choices. For instance, the abstract says "this manuscript reports" (line 28) that could be "this work reports"; the tense of verbs: "It takes almost 40min" (line 508) that should be "It took almost 40min", and the numbering of figures in the Supplemental material as Table S1, Figure S1.

Finally, the authors should describe the technical specifications of the computer used to simulate the RB model (at least, CPU model and RAM size).

Reviewer 3 ·

Basic reporting

The authors are clarified my concerns and I do think that the submission is almost ready for publication.

Yet I have few remarks that I think should be addressed in the final version.
-> As indicated in the rebuttal, the authors have not used KaDe because their model is containing some event-based interventions which are not covered by the tool yet. Maybe it should be explained in the paper, all the more so since event interventions do not belong to the pure-core of the rule-based paradigm.


* 440   Ambiguous molecularity.
-> I find the remark on ambiguous molecularity a bit misleading.
This is indeed an issue inherent to rule-based modeling. Since rules may describe biochemical complex only partially, it may happen that two patterns that are disconnected in a rule are indeed connected in the application of this rule. Giving the same kinetic to the case when they are connected and when they are disconnected in not realistic from a biophysical point of view.
This issue is indeed addressed both in Kappa and BNGL.
In Kappa by using uni/bimolecularity rates, in BNGL by using different agent separators (+,.) in the rules.

* 508  Simulation time -> Use KaSIm 4.1
-> The version of the Kappa simulator which has been used by the author is quite old.
Thus I am not so sure it makes sense to speak about computation time.
In particular, the data structures that is used to represent patterns of interest and their occurrences in the current state was not yet implemented.

e.g. see Pierre Boutillier, Thomas Ehrhard, Jean Krivine: Incremental Update for Graph Rewriting. ESOP 2017: 201-228

* line 463 - 467.

What "expanding a probability distribution" means is not so clear to me.
What is known is that if you take an event driven by an exponentially time-distributed clock and a another event driven by the conjunction of two exponentially time-distributed clocks, then the ratio between the total expected time and the standard variation is bigger in the case of the single event than in the case of the composite one (hence the total duration of the composite event is more predictable).

Experimental design

no comment

Validity of the findings

no comment

Additional comments

no comment

---

## Round 0.3 · accepted · Accept

The reviewers have no remarks. I endorse publication in current form.

Reviewer 1 ·

Basic reporting

I have no further comments. The authors addressed my concerns in their manuscript.

Experimental design

I have no further comments. The authors addressed my concern regarding the equivalence between concentration in a ODE-based model and copy number from an RBM.

Validity of the findings

I have no further comments regarding the validity of the findings

Additional comments

I have no additional comments.

Reviewer 3 ·

Basic reporting

My comments have been taken into account. The paper is ready for publication.

Experimental design

NA

Validity of the findings

NA

Additional comments

NA